# Distinct Hemostasis and Blood Composition in Spiny Mouse *Acomys cahirinus*

**DOI:** 10.3390/ijms252312867

**Published:** 2024-11-29

**Authors:** Nikita S. Filatov, Rafael R. Khismatullin, Airat I. Bilyalov, Alina I. Khabirova, Shakhnoza M. Salyakhutdinova, Roman V. Ursan, Roza N. Kasimova, Alina D. Peshkova, Insaf I. Gazizov, Elena I. Shagimardanova, Mary V. Woroncow, Andrey P. Kiyasov, Rustem I. Litvinov, Oleg A. Gusev

**Affiliations:** 1Institute of Fundamental Medicine and Biology, Kazan Federal University, 420008 Kazan, Russia; 2Loginov Moscow Clinical Scientific Center, 111123 Moscow, Russia; 3City Hospital № 11, 420127 Kazan, Russia; 4Department of Pharmacology, Perelman School of Medicine, University of Pennsylvania, Philadelphia, PA 19104, USA; alinapeshkova26@gmail.com; 5Institute for Regenerative Medicine, Lomonosov Moscow State University, 119991 Moscow, Russia; mworon@inbox.ru; 6Department of Cell and Developmental Biology, Perelman School of Medicine, University of Pennsylvania, Philadelphia, PA 19104, USA; litvinov@pennmedicine.upenn.edu; 7LIFT, Life Improvement by Future Technologies Institute, 121205 Moscow, Russia; 8Intractable Disease Research Center, Graduate School of Medicine, Juntendo University, Tokyo 113-8421, Japan

**Keywords:** *Acomys cahirinus*, hematological profile, fibrinogen concentration, haemostasis

## Abstract

The spiny mouse (*Acomys* species) is capable of scarless wound regeneration through largely yet unknown mechanisms. To investigate whether this capacity is related to peculiarities of the hemostatic system, we studied the blood of *Acomys cahirinus* in comparison to *Mus musculus* (Balb/c) to reveal differences in blood composition and clotting in both males and females. In response to surgical manipulations, blood clots formed in wounds of *Acomys* comprised a stronger hemostatic seal with reduced surgical bleeding in comparison with Balb/c. *Acomys* demonstrated notably shorter tail bleeding times and elevated clottable fibrinogen levels. Histological analysis revealed that clots from *Acomys* blood had densely packed fibrin-rich clots with pronounced fibrin segregation from erythrocytes. *Acomys* exhibited superior plasma clot stiffness as revealed with thromboelastography. The latter two characteristics are likely due to hyperfibrinogenemia. Light transmission platelet aggregometry demonstrated that ADP-induced platelet aggregates in *Acomys* males are stable, unlike the aggregates formed in the plasma of Balb/c undergoing progressive disaggregation over time. There were no apparent distinctions in platelet contractility and baseline expression of phosphatidylserine. Hematological profiling revealed a reduced erythrocytes count but increased mean corpuscular volume and hemoglobin content in *Acomys*. These results demonstrate the distinctive hemostatic potential of *Acomys cahirinus*, which may contribute to their remarkable regenerative capacity.

## 1. Introduction

In recent years, spiny mice, *Acomys cahirinus*, have emerged as a novel laboratory animal model for studying regeneration processes [1]. Spiny mice are the only mammal that regenerates tissues without fibrosis, showing restoration in skeletal muscle [2], repaired defects in skin flaps with associated structures such as hair [3,4], recovery of kidney function [5], and myocardial contractility after coronary artery ligation [6]. Additionally, Matias Santos et al. have documented the regeneration of a round defect in the auricle and elastic cartilage, a process that was initially deemed to have limited potential for recovery [7].

The exceptional recovery of *Acomys* in the face of pathologies rare in mouse species, including disorders related to coronary circulation or renal disease, could be an indirect result of the evolutionary development of its regenerative abilities. Curiously, atypical physiological manifestations such as menstruation [8] could also have played a role in shaping the *Acomys* biological profile. These new findings are growing, and our study demonstrates marked changes in hemostasis in spiny mice, possibly shaped by a variety of adverse factors and evolutionary refinement of regenerative capacity. The results obtained suggest a complex interaction of unique physiological manifestations with habitat-typical lesions and the blood coagulation potential in spiny mice. Clearly, the evolutionary pathway of *Acomys* involves a multifaceted response to various challenges, providing a glimpse into the complex process of its adaptive and regenerative abilities.

In this report, we present the results obtained during the comparative analysis of the hemostatic system and blood composition in *Acomys* versus *Mus musculus* (Balb/c), including tail bleeding time and fibrinogen concentration, along with a number of other hemostatic laboratory tests and hematological profile. The findings of this study shed light on the physiological differences between spiny mice and other rodents and support the important role of hemostasis in wound healing and regeneration.

## 2. Results

*Acomys cahirinus* have improved hemostasis in vivo associated with hyperfibrinogenemia and dense fibrin-rich blood plasma clots.

During surgical interventions conducted on *Acomys cahirinus*, distinct variations in both bleeding duration and blood loss were conspicuously observed in comparison to Balb/c, irrespective of the size of the surgical injury. Notably, these distinctions manifested as a reduced necessity for interventions to attain complete hemostasis at the surgical site, particularly when interacting with medium-size arteries or during the excision of substantial, well-vascularized tissue masses. In most cases, full cessation of bleeding in *Acomys* could be effectively achieved by just clamping the artery, whereas Balb/c necessitated supplementary measures to achieve hemostasis. Moreover, a significant contrast was noted in the incidence of postoperative complications attributable to blood loss between the two species of mice. Notably, none of the spiny mice succumbed to postoperative hemorrhage following surgical procedures involving arterial manipulation, in a stark contrast to the Balb/c, signifying the notable divergence in their hemostatic responses.

To confirm these observations and quantitatively assess the overall hemostatic potential, a tail bleeding test was conducted, which revealed a significant difference in the duration of bleeding between *Acomys* and Balb/c (Figure 1A) irrespective of gender. In the tests performed, spiny mice showed about 2-fold shorter median bleeding time, which is consistent with the reduced propensity to bleeding in *Acomys* observed during surgical manipulations. 

These in vivo observations led us to a hypothesis that there must be substantial variations in the functionality of the hemostatic system between Balb/c and *Acomys*. The experiments described below have confirmed this presumption.

The levels of clottable fibrinogen in blood plasma in the two mouse species turned out to be significantly different (Figure 1B). The mean fibrinogen concentration was substantially higher in spiny mice compared to Balb/c (Figure 1B), which corresponded to the shorter tail bleeding time in *Acomys* (Figure 1A). This finding suggests that blood clots formed in *Acomys* may have the structure and properties promoting hemostasis and wound healing.

To see if the revealed hyperfibrinogenemia in *Acomys cahirinus* affects clot structure and composition, freshly prepared clots from the blood of both species were fixed, sectioned, stained, and analyzed microscopically. The visual inspection revealed significant morphological distinctions. 

In Figure 2A,C, representative images of a clot formed from the blood of *Acomys cahirinus* are shown, which clearly demonstrate densification/compaction of the major structural elements, namely erythrocytes and fibrin. In addition, a substantial predominance of the fibrin volume fraction is observed in *Acomys* compared to the blood clots of Balb/c (Figure 2B,D). Notably, in the *Acomys* blood clots, fibrin was clustered and formed densely packed aggregates and clumps, clearly separated from erythrocytes (Figure 2E) which was in contrast with the clots from Balb/c (Figure 2F).

The images on the right (Figure 2B,D), representing a typical blood clot from a Balb/c mouse, show loose packing of the structural elements. In contrast with the blood clots from *Acomys*, fibrin is relatively sparse, dispersed over a larger area of the section, and is often colocalized with erythrocytes, resulting in many erythrocytes staining pink rather than typical yellow (Picro–Mallory stain). Accordingly, fibrin mainly does not form large and dense agglomerates, which distinguishes these clots (Figure 2F) from those formed from the blood of *Acomys cahirinus* (Figure 2E).

### 2.1. Clots from the Blood of Acomys cahirinus Are Stiffer

To assess hemostasis using an integral test, we employed thromboelastography (TEG), a method that monitors blood clotting based on the dynamic elastic properties of a newly formed blood clot. TEG was performed in whole citrated blood and platelet-free plasma (PFP) samples from both genders of *Acomys cahirinus* and Balb/c (Figure 3).

In whole blood, the rate of thrombin generation reflected by the onset of fibrin formation (parameter *R*) is accelerated in Balb/c compared to spiny mice in whole blood samples (Figure 3A). Accordingly, the rate of fibrin polymerization reflected by the alpha angle (α) was significantly higher in Balb/c (Figure 3B). The final clot stiffness (*G*, shear elastic modulus) was indistinguishable between the mouse species in males, but the clots were stiffer in Balb/c females (Figure 3C).

In PFP samples (Figure 3D), the reaction time (*R*) was also prolonged in *Acomys* samples as in whole blood, while the difference in rate of fibrin polymerization (alpha angle) was statistically insignificant (Figure 3E). Unlike in whole blood, the clot stiffness reflected by shear elastic modulus (*G*) in the plasma samples obtained from spiny mice was significantly higher in males and followed the same trend in females (Figure 3F), which is consistent with hyperfibrinogenemia (Figure 1B) and the dense compact blood plasma clots (Figure 2) observed in *Acomys*.

In line with the increased plasma clot stiffness revealed by TEG, the clots obtained from the blood of *Acomys* were apparently stiffer and less deformable when cut on the microtome during sample preparation for histology, confirming that they were firmer.

### 2.2. Peculiarities of Platelet Function in Acomys cahirinus

Clot contraction is driven mainly by activated platelets, but it depends also on the cellular and protein composition of the blood. Therefore, the blood clot contraction kinetics can be used to assess platelet contractility as a part of the overall hemostatic potential [9].

The final extent of contraction and average velocity were significantly increased in whole blood samples obtained from male spiny mice (Figure 4A,B), while in females, the difference was the opposite, i.e., the final extent of contraction and average velocity were smaller in female spiny mice versus Balb/c. The latter regularity was conformed in PRP samples; namely, the final extent of contraction was significantly higher in Balb/c of both genders and the average velocity was significantly higher in Balb/c females (Figure 4C,D).

Thereafter, to study platelet reactivity, we applied light transmission aggregometry (LTA) in response to stimulation with ADP (adenosine diphosphate). According to the results of LTA, no statistically significant difference in the maximum extent of platelet aggregation was observed between *Acomys cahirinus* and Balb/c, irrespective of gender (Figure 5). However, there was a difference in the kinetics of platelet aggregation in males (Figure 5A). Platelets from the *Acomys cahirinus* sample reached the averaged maximum extent of aggregation in about 7 min, whereas the Balb/c sample reached the averaged peak 3 min after the initiation of aggregation with ADP. However, at a ~12-min time point, the average extent of aggregation of platelets obtained from spiny mice became significantly higher by ~14% (*p* = 0.034) due to progressive platelet disaggregation in the Balb/c ADP-activated PRP samples. This significant difference in male mice was maintained through the entire observation period until 30 min (Figure 5A), while there was no difference revealed in the final degree of aggregation between female mice of both species (Figure 5B).

To evaluate the procoagulant activity of the most abundant blood cells, the level of phosphatidylserine expression on the surface of platelets and erythrocytes was quantified using flow cytometry. The expression of phosphatidylserine on the surface of platelets and erythrocytes obtained from *Acomys cahirinus* and Balb/c was found to be low and statistically insignificant both in males and females (*p* > 0.05 for all), suggesting comparable background procoagulant activity of quiescent blood cells.

### 2.3. Variations in Cellular Blood Composition Between Acomys and Balb/c

Hematological parameters are shown in Table 1. The mean count of red blood cells (RBCs) in the blood from both male and female *Acomys cahirinus* was reduced compared to Balb/c, while the median mean cell volume (MCV), the mean content of hemoglobin per one erythrocyte (MCH), and the mean corpuscular hemoglobin concentration (MCHC) were significantly higher. Hemoglobin concentration and hematocrit were found to be elevated only in the blood samples from female Balb/c mice. Platelet counts did not exhibit a statistically significant difference between the samples obtained from both genders of *Acomys*; however, mean platelet volumes (MPVs) were higher in *Acomys*, regardless of gender. The total white blood cell (WBC) count was higher in *Acomys* compared to Balb/c mice solely in males. Concurrently, the samples obtained from male *Acomys* showed an elevated lymphocyte count and a reduced neutrophil count in comparison to the Balb/c samples. Based on the results of the leukocyte counts, the neutrophil-to-lymphocyte ratio (NLR) was calculated, which was higher in the *Acomys* samples (Table 1).

## 3. Discussion

*Acomys cahirinus* are remarkable for the ability to regenerate tissues efficiently without a scar, but the mechanisms underlying this unique feature have not yet been elucidated. Given the major role that hemostasis plays in wound healing [10], the unique regenerative ability may be related to the peculiarities of blood clotting and platelets in *Acomys*. It is conceivable that systemic hypercoagulability and stiff blood clots may relate to the unique ability of *Acomys* to readily shed a considerable portion of their dorsal skin as a defense mechanism against predators. To test this hypothesis, we assessed the hemostatic potential in *Acomys cahirinus* compared to *Mus musculus* (Balb/c).

One of the main findings is the shorter bleeding time in *Acomys* compared to Balb/c (Figure 1A). To elucidate the possible mechanisms of this observation, the concentration of clottable fibrinogen in the plasma of *Acomys* was studied, which turned out to be on average about 20–40% higher than in Balb/c (Figure 1B). Notably, the elevated blood levels of fibrinogen in male *Acomys* are higher than the normal levels reported for other mouse strains [11,12]. The revealed hyperfibrinogenemia in *Acomys* is consistent with the reduced bleeding time as fibrinogen level in blood is a major determinant of hemostatic potential [13].

The elevated fibrinogen concentration in the blood is reflected by the morphological characteristics of blood clots obtained from *Acomys cahirinus* compared to Balb/c, such that the *Acomys* clots have much denser structure with the prevailing volume fraction of fibrin (Figure 2). Accordingly, blood clots from male spiny mice are much stiffer, as reflected by the increased shear elastic modulus measured with TEG (Figure 3F) and observed empirically while cutting the clots on the microtome in preparation for histology. The same trend towards stiffer plasma clots in *Acomys* versus Balb/c mice is seen in females (Figure 3F), while in whole blood clots, the result is the opposite (Figure 3C), likely reflecting the difference in hematocrit and bulk mechanical properties of RBCs that overcome the effect of fibrin in plasma clots without RBCs. The increased density and stiffness of the *Acomys* blood plasma clots is apparently due to a larger amount of fibrin formed, which underwent covalent cross-linking with Factor XIIIa and perhaps additional compaction driven by contractile activated platelets bound to fibrin via the integrin receptor αIIbβ3. Clot contraction causes densification and stiffening of fibrin within a blood clot [14].

Blood clot contraction is a multifactorial process involving many blood components that can modulate the extent and velocity of contraction, sometimes overcoming the main contribution from activated and contractile platelets. Therefore, the enhanced contraction in whole blood clots seen in male *Acomys cahirinus* (Figure 4A,B) is likely explained by reduced RBC counts known to be inversely related to the rate and extent of clot contraction [15]. In contrast, a reduction in the extent of clot contraction observed in female whole blood and PRP samples from spiny mice of both genders may be caused by hyperfibrinogenemia and high fibrin content with a remarkable mechanical resilience [15]. Given the same platelet counts in both mouse species studied (Table 1), the overall results on clot contraction support the conclusion that there is no substantial difference in platelet contractility between *Acomys* and Balb/c. Hematological profiling demonstrated that platelets in *Acomys* on average were significantly larger than the platelets in Balb/c. However, it remains unclear whether the larger platelets observed in *Acomys* may affect the overall platelet contractility. Although contraction of blood clots remains an important mechanism in hemostasis and thrombosis [9], the interplay between blood clot contraction and wound healing remains underestimated and poorly understood [16].

An interesting and unexpected finding is that *Acomys* platelets in males form much more stable aggregates, although their initial response to stimulation with ADP is somewhat slower than in Balb/c (Figure 5A). As soon as platelet aggregation is mediated by the bridging interactions between bivalent fibrinogen molecules and active integrin αIIbβ3 on the surface of adjacent platelets [17], formation of mechanically strong and stable aggregates may be another physiological consequence of hyperfibrinogenemia. In relation to hemostasis, the stability of platelet aggregates may contribute to mechanical and lytic resistance and overall sustainability of blood clots formed at the site of injury.

Despite the overall evidence for hypercoagulability and enhanced hemostasis in *Acomys* compared to Balb/c, the kinetic parameters of TEG look contradictory. Both in whole blood and PRP, the longer time to initiate clotting (reaction time, *R*) and decreased fibrin polymerization rate (α angle) in *Acomys* are indicative of impaired or delayed blood clotting. However, this seeming contradiction may be attributed to the high fibrinogen level in the blood of *Acomys*, because fibrinogen at high concentrations is known to partially inhibit fibrin polymerization in a competitive manner and delay clot formation [18].

The hematologic profile of spiny mice has shown that *Acomys* have relatively small RBC counts, which is possibly compensated for by an increase in MCV, MCH, and MCHC. The cause and effects of these variations in cellular blood composition remain to be further investigated. It is important to note that hematologic profiles in mice vary significantly across different studies reported in the literature [19,20,21,22], including hematocrit (HCT), mean corpuscular volume (MCV), and white blood cell (WBC) values. The differences observed in parameters such as a higher hemoglobin concentration and hematocrit among female Balb/c mice can be attributed to the fact that *Acomys cahirinus*, the only rodent species with a menstrual cycle, exhibits uterine bleeding associated with anemia [23]. This physiological feature may provide an explanation for the observed gender-related variability in some hemostatic parameters, such as the extent of clot contraction and platelet aggregation, which is in line with the gender-based hormonal profiles and hormone fluctuations that may affect the hemostatic potential [24,25,26,27]. What is really important is that the main hemostatic distinctions of *Acomys cahirinus* (short bleeding time, elevated fibrinogen level, high density and stiffness of fibrin-rich clots) remain independent of gender and reflect fundamental and unique biological features of spiny mice.

Significant differences between *Acomys cahirinus* and Balb/c were also found in the total WBC counts and leukocyte types, and these variations may be related to the elevated fibrinogen level as a marker of inflammation [28]. Therefore, it can be assumed that spiny mice, unlike Balb/c, may be in a state of persistent chronic inflammation. However, the results of the hematological profiling, including the neutrophil-to-lymphocyte ratio (NLR), which reflects the state of both innate and adaptive immunity [29], suggest that the overall reactivity of the immune system is approximately the same in the two species of mice studied.

The most remarkable finding in this study is a substantially elevated level of fibrinogen in the blood of *Acomys cahirinus* irrespective of gender. Since fibrinogen has a dual function as a protein involved in hemostasis and immunity, hyperfibrinogenemia may reflect changes in the degree of inflammatory response, which is a significant determinant of the outcome of tissue regeneration [28,30]. Fibrinogen can influence immune cells through the integrins αXβ2 (CD11c/CD18, p150.95) and αMβ2 (CD11b/CD18, Mac-1) [31] on their surface, thus regulating monocyte adhesion and activity [32]. However, despite elevated fibrinogen levels, the immune response to injury is reduced in *Acomys*, which represents a significant distinguishing characteristic when compared to other rodent models. Fibrin induces anti-inflammatory polarization in both mouse and human macrophages, shifting them from a pro-inflammatory M1 phenotype to an anti-inflammatory M2 phenotype [33]. Studies have also indicated that fibrin can modulate macrophage phenotype behavior, potentially providing an immunomodulatory strategy for tissue healing and regeneration [34]. According to some studies, *Acomys* exhibits an increased number of anti-inflammatory M2 macrophages in regenerating tissues [1,35]. Besides this, macrophages play an important role in regeneration, removing debris, producing cytokines and growth factors, and controlling and limiting tissue growth after completion of repair [36,37]. Additionally, fibrin can directly impact angiogenesis, allowing endothelial cells to infiltrate the matrix and form new capillary structures [38,39]. All of these aspects appear to be directly related to *Acomys cahirinus*’s heightened regenerative capacity, and investigation of the impact of macrophages may prove invaluable in elucidating the underlying mechanisms of this distinctive capacity.

In addition to variations in systemic inflammation, a degree of local inflammation at the site of injury may play a crucial role in this unique regenerative ability. Hyperfibrinogenemia in *Acomys* leads to deposition of a larger amount of dense fibrin, which is additionally compacted during enhanced clot contraction (Figure 2 and Figure 3). Such fibrin-rich and dense blood clots can form an impermeable seal that impedes penetration and accumulation of inflammatory cells, neutrophils, and monocytes, thus impeding the local inflammation. This reduced number of immune cells in the wound leads to the lower local concentrations of inflammatory mediators, including chemotactic cytokines, followed by decreased migration and low activation of fibroblasts synthesizing collagen, the major component of a scar [40,41,42,43,44]. To test this presumptive explanation, additional studies examining the local cellular composition of the wound during healing dynamics are necessary.

In summary, we have revealed clear distinctions in the functional state of the hemostatic system in *Acomys cahirinus* compared to Balb/c. Due to the increased concentration of fibrinogen in the blood of *Acomys cahirinus*, the tail bleeding time is shorter; blood clots obtained from the blood of spiny mice are denser and stiffer, causing formation of a stronger hemostatic seal and reduction of surgical bleeding in comparison with Balb/c. Contraction of whole blood clots in male spiny mice is stronger and faster than in Balb/c, likely due to reduced RBC counts. On the contrary, contraction of whole blood clots in female mice and of plasma clots in both genders is weaker in *Acomys* due to increased mechanical resilience of the larger amount of fibrin associated with hyperfibrinogenemia. ADP-induced platelet aggregates in male *Acomys* are stable, unlike the aggregates formed in the PRP of Balb/c that undergo progressive disaggregation over time. The distinct hemostasis in *Acomys cahirinus* and Balb/c highlights significant physiological adaptations in *Acomys*, characterized by a trend towards hypercoagulability. The results obtained suggest that the special features of the hemostatic system underly the enhanced tissue regeneration in spiny mice. This research provides a foundation for exploring the interplay between hemostasis and regeneration, with important potential medical implications.

## 4. Materials and Methods

### 4.1. Animals and Ethical Issues

The experimental procedures were conducted on male and female *Acomys cahirinus* and Balb/c (Krolinfo Ltd., Moscow, Russia) at 11–12 weeks of age. No less than 5 animals of each mouse species and gender were used in the particular experiments. “Zoletil 100” at a dosage of 7 mg per 1 kg of body weight was used as anesthesia in all cases. Ethical approval for all animal studies was obtained from the local Ethics Committee of the Institute of Fundamental Medicine and Biology, Kazan (Volga region) Federal University, under extract No. 40 dated 9 March 2023.

### 4.2. Blood Collection and Processing

Blood was collected via puncture of the left ventricle of the heart in tubes (Eppendorf SE, Hamburg, Germany) containing 3.2% sodium citrate at a volume ratio of 9:1. Platelet-rich plasma (PRP) was obtained by centrifugation of citrated blood at 200× *g* for 5 min at room temperature and used for flow cytometry, light transmission aggregation of platelets, and the clot contraction assay. To obtain platelet-free plasma (PFP), PRP was centrifuged at 10,000× *g* for 10 min, and used for TEG and APTT. All samples were used within 4 h after blood collection.

### 4.3. Tail Bleeding Test

The animals were positioned in a supine posture on a 37 °C heated table, and the level for tail tip amputation was marked. A pointed scalpel was used to amputate the tail tip at the predetermined level, resulting in a stump 2 mm in diameter. Immediately after the amputation, the first drop of blood was removed from the stump’s surface using filter paper. Subsequently, the tail was immersed in a 50 mL glass container filled with preheated 0.9% sodium chloride solution at 37 °C. The solution’s temperature was maintained using a water bath, and the tail stump was situated approximately 2–3 cm below the animal’s body level. The bleeding time was measured using a stopwatch until complete cessation of bleeding. Following the procedure, the tail stump underwent antiseptic treatment, and the animal was returned to its original housing conditions.

### 4.4. Histological Examination of Blood Clots

The animals were placed in the supine position on a table heated to 37 °C, and the level of tail amputation was marked 3 cm from its base. After amputation, blood was drawn from the obtained stump into 1.5 mL Eppendorf tubes pre-lubricated with a detergent (4% Triton X-100 in saline) to prevent sticking of the clot to the walls.

The obtained non-stabilized blood samples were incubated at 37 °C for 5 min for clotting. After clot formation, they were fixed by adding buffered 10% formalin (4% formaldehyde) as a fixative. After fixation at room temperature for 24 h, the clots were subjected to the standard protocol of paraffin embedding, after which 3–4 µm thick sections were made on a rotary microtome. The obtained sections were stained using the Picro–Mallory staining method according to the standard protocol with the ErgoProduction kit (Saint-Petersburg, Russia).

### 4.5. Determination of Clottable Fibrinogen in Plasma

A sample of citrated platelet-free plasma (200 µL) was re-calcified by adding 2 µL of 0.2 M calcium chloride (2 mM final) and blood clotting was initiated by adding 25 µL of 40 U/mL human thrombin (5 U/mL final) at 37 °C. After 30 min of incubation, the clot was washed with saline (3 times × 30 min) and left overnight in saline to wash out plasma proteins. The clot was blotted with filter paper and dissolved in 400 μL of 0.25 M NaOH by heating in boiling water for 5 min. The protein absorbance was determined at 280 nm in a Nanodrop Lite spectrophotometer (ThermoFisher Scientific, Waltham, MA, USA) and converted into protein concentration. The specific absorption coefficient for fibrin(ogen) was equal to 1.51 for 1 mg/mL in a 1 cm cuvette and the calculations were performed using the following formula: C = (A_280_/1.51) × T × P), where C is fibrin concentration, A_280_ is the measured absorbance, T is a beam path corresponding to width of the cuvette (1 cm), and P is correction for the sample dilution. This protocol allowed the determination of a clottable fibrinogen level in the blood plasma.

### 4.6. Thromboelastography (TEG)

TEG was conducted on whole blood and PFP samples. To a total volume of 350 µL blood or PFP, 15 µL of kaolin (0.01 ng/mL final) was added. After incubation for 5 min at 37 °C in a thermostat, 340 µL of the activated sample was transferred to a cuvette already containing 20 µL of calcium chloride (final concentration 20 mM) and recording was initiated in a hemostasis analyzer TEG 5000 (Haemoscope Corporation, Niles, IL, USA). The following parameters were extracted from a thromboelastogram: reaction time (R) reflecting the time needed for thrombin generation and initial fibrin formation, angle (α) showing a fibrin polymerization rate, maximum amplitude (MA) depending on the clot stiffness, and shear elastic modulus G, reflecting the actual clot strength. G is derived from MA using the following formula provided by the manufacturer of TEG 5000: G (dyn/cm^2^) = (5000 × MA/(100 − MA)) (Figure 6).

### 4.7. Clot Contraction Assay

The kinetics of blood clot formation were determined using a method based on the optical registration of blood clot size over time, employing a Thrombodynamics Analyser (HemaCore Ltd., Moscow, Russia). Prior to the determination, a dual-channel plastic measuring cuvette provided with the instrument was pre-lubricated with a 4% Triton X-100 solution in 150 mM NaCl (for whole blood) and with 1% pluronic (for PRP). This was done to prevent fibrin from sticking to the cuvette walls and to allow for unconstrained contraction of a blood clot. To 200 µL of citrated blood, 2 mM CaCl_2_ (final concentration) was added. Subsequently, 1.5 U/mL human thrombin (final concentration) was introduced to initiate the clotting and platelet activation. For the PRP sample, 10 mM CaCl_2_ (final concentration) was added to 200 µL of PRP, then thrombin was added at a final concentration of 1.5 U/mL. Afterwards, 80 μL of the activated sample was rapidly transferred into a transparent cuvette preheated to 37 °C in a thermostated chamber. The photoregistration of blood clot size was conducted automatically at 15 s intervals over 20 min. The serial images were processed, and a kinetic curve was built, from which the following parameters were determined: final extent of contraction, lag time, area under curve, and average velocity. The final extent of contraction defines the maximum level of clot contraction achieved in 20 min. This parameter is estimated as the reduction in a relative clot size from the initial stage to the final contraction (%). Average velocity is the rate of clot contraction per unit time.

### 4.8. Flow Cytometry of Erythrocytes and Platelets

To determine the phosphatidylserine expression level on the surface of erythrocytes and platelets, washed erythrocytes and PRP samples were used. Erythrocytes were pelleted by centrifugation of citrated blood at 200× *g* for 5 min and washed by resuspending in Tyrode’s buffer (4 mM HEPES, 135 mM NaCl, 2.7 mM KCl, 2.4 mM MgCl_2_, 5.6 mM D-glucose, 3.3 mM NaH_2_PO_4_, 0.35 mg/mL bovine serum albumin, pH 7.4) three times. Surface-associated phosphatidylserine of platelets and erythrocytes was determined using fluorescein isothiocyanate (FITC)-labeled annexin V (BioLegend, San Diego, CA, USA). A total of 1 μL of PRP or erythrocyte sample was diluted with 49 μL Tyrode’s buffer, mixed with labeled annexin V (final concentration 1×), and incubated for 15 min at room temperature in the dark. A total of 50 μL of diluted and labeled erythrocytes/PRP were mixed with 350 μL Ca^2+^binding buffer (10 mM HEPES, 140 mM NaCl, 2.5 mM CaCl_2_, pH 7.4) and analyzed on a FacsCalibur flow cytometer (Becton Dickinson, Franklin Lakes, NJ, USA). For each sample, 5000 events of platelets and erythrocytes were collected. Data processing was performed using FlowJo vX.0.7. software (Becton Dickinson, Franklin Lakes, USA). The fraction of erythrocytes or platelets expressing phosphatidylserine was determined as annexin V-FITC-positive signals in the erythrocyte or platelet gate.

### 4.9. ADP-Induced Platelet Aggregation

Platelet aggregation was studied in 100 μL of PRP diluted with 100 μL of Tyrode’s buffer. Light transmission aggregometry (LTA) was employed using a Biola LA230-2 instrument (Biola, Moscow, Russia). The PRP sample was placed in an aggregation cuvette pre-heated to 37 °C. Thereafter, 20 μM (final concentration) of ADP (adenosine diphosphate) was added as a platelet agonist, and the aggregation was recorded for 30 min. The resulting response was registered as an aggregation curve generated by the software integrated into the aggregometer, based on the light transmission of the sample.

### 4.10. Complete Blood Count

For hematological analysis, blood was drawn into vacuum mini-tubes containing K_3_-EDTA (Weihai Hongyu Medical Devices Co. Ltd., Weihai, China) at a volume of 0.5 mL per tube. Subsequently, the samples were analyzed using an automatic hematological analyzer Urit-5380 Vet (URIT Medical Electronic Group Co., Shenzhen, China) calibrated according to the manufacturer’s instructions. 

### 4.11. Statistical Analysis

The numerical data were analyzed using the software package GraphPad Prism version 9.0. The normality of the distribution was assessed using the Shapiro–Wilk, D’Agostino–Pearson, and Kolmogorov–Smirnov criteria. Two-sided *t*-test (parametric) or Mann–Whitney U-test (nonparametric) were utilized to determine the statistical significance of the observed differences between the data arrays of the two groups of data. Multi-group analysis was performed using ANOVA to analyze and compare platelet aggregatograms. Statistical significance was considered at *p* < 0.05.

## 5. Conclusions

The results obtained indicate substantial peculiarities in the hemostatic system and blood composition in *Acomys cahirinus*. These observations are important for the fundamental knowledge about the physiology of spiny mice, including the ability to enhance tissue regeneration; however, the mechanistic relationships of hemostasis and regeneration require more detailed and in-depth studies. *Acomys cahirinus* is a promising candidate for the role of a new animal model for fundamental studies of the relationship between hemostatic capacity and the distinctive ability to regenerate tissue following damage, as well as the processes of physiological adaptation of the hemostasis system to environmental conditions.

## Figures and Tables

**Figure 1 ijms-25-12867-f001:**
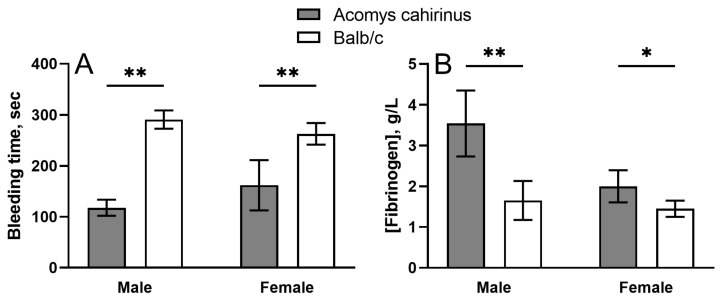
Results of the tail bleeding assay (**A**) and levels of clottable fibrinogen in blood plasma (**B**) of male and female *Acomys cahirinus* mice versus Balb/c. The results are presented as mean ± SD (n = 5 for each species and gender group). * *p* < 0.05; ** *p* < 0.01.

**Figure 2 ijms-25-12867-f002:**
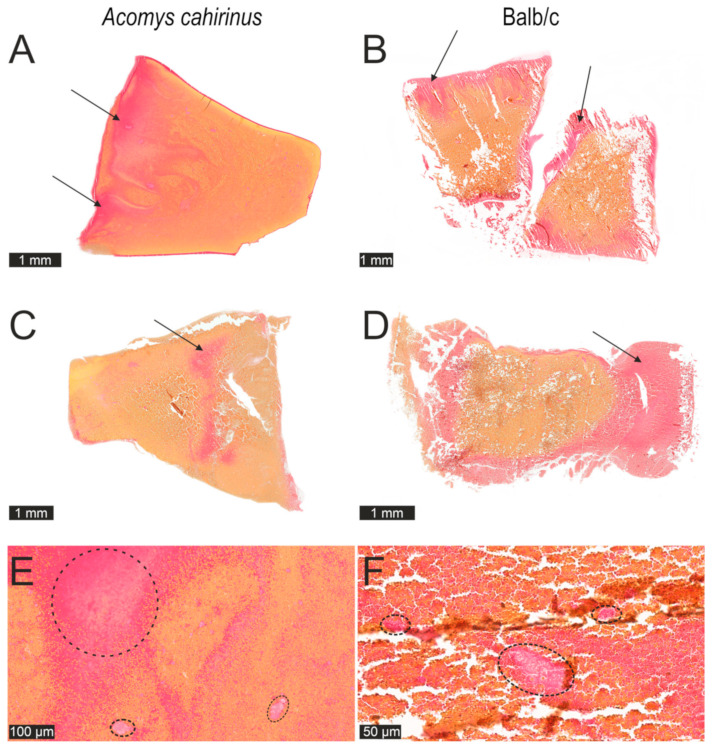
Representative histological images of blood clots from *Acomys cahirinus* (**A**,**C**) and Balb/c (**B**,**D**) (n = 3 for each). Arrows indicate *red* fibrin in the clot. Erythrocytes are *yellow*. (**E**) Characteristic dense *red* fibrin accumulations within the *Acomys cahirinus* blood clot. Multiple giant densely packed clusters and fibrous structures of fibrin are visualized (within dashed ovals), mostly separated from *yellow* erythrocytes. (**F**) Typical loose distribution of fibrin within the Balb/c blood clot. A significant predominance of loosely arranged fibrin (*red*), colocalized with erythrocytes (stained *pink*, not *yellow* as in the *Acomys* clot). Single islets of fibrin aggregates (marked by dashed ovals) were observed. Picro–Mallory stain. Magnification 25× (**A**–**D**) and 200× (**E**,**F**).

**Figure 3 ijms-25-12867-f003:**
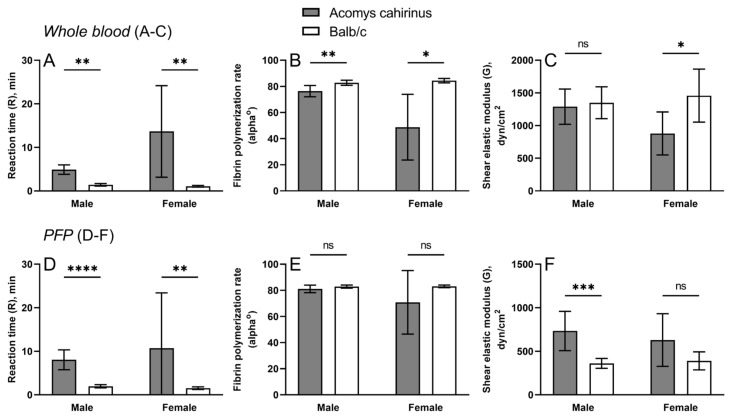
TEG parameters of whole blood (**A**–**C**) and PFP (**D**–**F**) in male and female *Acomys cahirinus* and Balb/c: *R* (reaction time), α-angle (fibrin polymerization rate), and *G* (shear elastic modulus). n = 6 for paired whole blood samples and n = 9 for PFP samples from males; n = 5 for paired blood and PFP samples in females. The results are presented as mean ± SD. * *p* < 0.05, ** *p* < 0.01, *** *p* < 0.001, **** *p* < 0.0001. ns—not significant.

**Figure 4 ijms-25-12867-f004:**
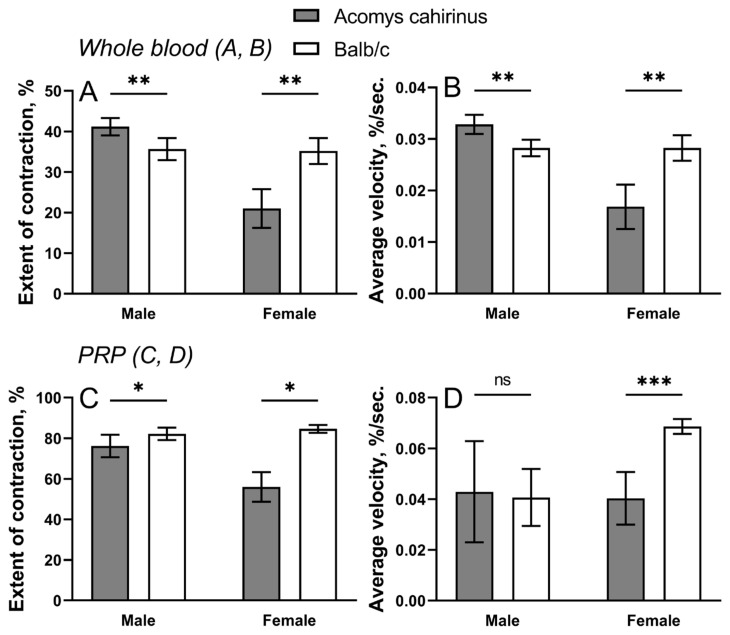
Parameters of clot contraction in whole blood (**A**,**B**) and in PRP (**C**,**D**) from *Acomys cahirinus* and Balb/c males and females. n = 6 for paired whole blood samples and n = 5 for PRP samples from males; n = 5 for paired blood and PRP samples from females. The results are presented as mean ± SD. * *p* < 0.05, ** *p* < 0.01, *** *p* < 0.001. ns—not significant.

**Figure 5 ijms-25-12867-f005:**
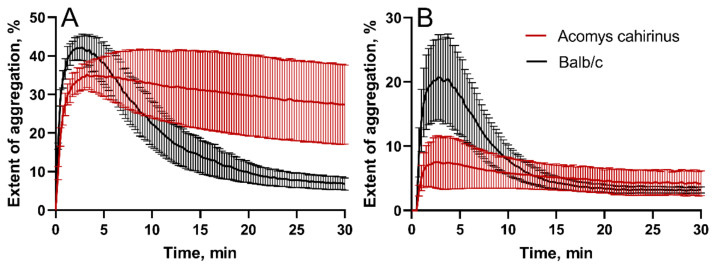
Averaged light transmission aggregograms recorded after ADP-induced platelet aggregation in PRP from male (**A**) and female (**B**) *Acomys cahirinus* and Balb/c (n = 5 for each species and gender group). Two-way ANOVA test in (**A**): *p* < 0.05 at 12 min, *p* < 0.01 at 15 min, *p* < 0.001 at 20–30 min.

**Figure 6 ijms-25-12867-f006:**
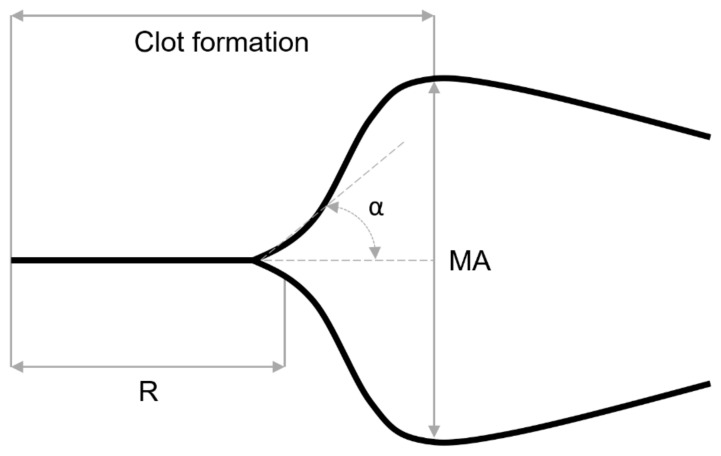
A characteristic thromboelastogram and the key parameters of clot formation: *R*—reaction time, α—alpha angle, *MA*—maximum amplitude.

**Table 1 ijms-25-12867-t001:** Hematological profiles of *Acomys cahirinus* and Balb/c (n = 5 for each mouse species and gender). The results are presented as mean ± SD. n.s.—not significant.

Hematological Parameters	*Acomys cahirinus*		Balb/c		*p* Value	
	Male	Female	Male	Female	Male	Female
Red blood cells (RBCs), ×10^12^/L	8.8 ± 1.1	8.0 ± 0.5	11.0 ± 1.5	11.6 ± 0.8	0.02	0.008
Hemoglobin, g/L	149.4 ± 22.6	139.4 ± 3.3	145 ± 21.6	163.4 ± 8	n.s.	0.01
Hematocrit, %	62.2 ± 9.2	57.9 ± 3	63.4 ± 9.3	73.2 ± 5	n.s.	0.008
Mean corpuscular volume (MCV), fL	70.6 ± 2.3	72.6 ± 2.1	57.6 ± 2.2	63.2 ± 1.5	0.0006	0.008
Mean content of hemoglobin per one erythrocyte (MCH), pg	16.9 ± 0.9	17.4 ± 0.9	13.1 ± 0.7	14 ± 0.7	0.002	0.009
Mean corpuscular hemoglobin concentration (MCHC), g/L	240.3 ± 4.3	241 ± 8.5	228 ± 3.8	223 ± 7.7	0.002	0.03
Platelets, ×10^9^/L	415 ± 51.3	391.2 ± 49.6	437.6 ± 32.8	425.4 ± 38	n.s.	n.s.
Mean platelet volume (MPV), fL	10.8 ± 0.3	11 ± 0.2	9.2 ± 0.2	9.6 ± 0.1	0.002	0.01
White blood cells (WBCs), ×10^9^/L	3.2 ± 1.2	2.15 ± 1.3	1.3 ± 1.3	1.7 ± 0.7	0.03	n.s.
Neutrophils, %	17.7 ± 9.6	16.4 ± 11	33.1 ± 9.7	16.8 ± 5.8	0.03	n.s.
Lymphocytes, %	74.9 ± 9.5	83.6 ± 11	60.6 ± 10.6	83.2 ± 5.7	0.046	n.s.
Monocytes, %	4.6 ± 2.8	0	4.6 ± 4	0	-	n.s.
Neutrophil-to-lymphocyte ratio (NLR)	0.3 ± 0.2	0.2 ± 0.2	0.6 ± 0.3	0.2 ± 0.1	0.03	n.s.

## Data Availability

The raw data used to create the figures in this article is available upon request from the corresponding author.

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
