# Peer review of "Distinct Hemostasis and Blood Composition in Spiny Mouse *Acomys cahirinus"

_ijms, 2024, doi:10.3390/ijms252312867_

Round 1
Reviewer 1 Report
Comments and Suggestions for Authors
Filatov et al performed haemostatic and hematological measurements comparing Acomys cahirinus (Spiny mouse) with normal BALB/c mice (Mus musculus), and found a significantly-reduced bleeding time with a 40% higher level of blood fibrinogen in Acomys mice. This study provides important information to the existing literature helping to establish Acomys mice as a new model for the study of human physiology and pathology, especially in the area of regenerative medicine. Authors may wish to add more observations to further improve the report.
Specific comments
Data presented in Figures 1 and 2 can be combined to reduce total number of figures in the manuscript. More importantly, the argument authors made at line 348-350 in methods regarding animal gender selection is not valid. It is very important to know whether any specific change in blood counts and haemostatic measurements is gender specific, especially when Acomys mice are gaining popularity as model animals for the study of human pathology/disease therapy. Authors need to make new measurements comparing female Acomys and BALB/c mice and add the new data to current figures.
Blood collection and complete blood count were briefly described in the methods section. However, data presented in Table 1 seemed to be somewhat out of the ordinary: HCT and MCV values were too high while WBC value was too low for Balab/c mice. Were the Balb/c mice normal/healthy? Did authors make good calibration to the CBC analyzer before actual sample measurements? It is recommended that authors calibrate their CBC analyzer with good Mus musculus standards before future sample analyses.
The discussion at lines 269-277 regarding large platelets from Acomys mice were inaccurate or invalid since authors did not present any data concerning platelet maturity (reticular platelets) in the current study. The related discussion is baseless. According to data presented in Table 1, MPV was 10.8 for Acomys mice and was 9.2 (this MPV value is too large for Balab/c mice under normal circumstance), which represents a 17% higher MPV in Acomys mice. Since Acomys mice are generally larger and live longer than Balb/c mice, having slightly higher MPV and MCV should be regarded as normal, unless authors can provide any specific evidence showing that the platelets from Acomys mice were pre-mature.
The Spiny mouse’s unique characteristic of “scarless wound healing” relative to normal Mus musculus mice has been the focus of the study. However, data obtained from current study indicated that Spiny mice are hyperfibriogenemic, with specific data showing in Figure 3E with larger fibrin aggregates in blood clots. Authors need to make a better connection/explanation between the “scarless” nature of Acomys mice and current observations.
Author Response
We would like to thank the Reviewer for their thorough review of our manuscript. To address the Reviewer concerns and excellent suggestions, we have performed additional experiments and revised the text based on the Reviewer comments. We think that these changes have substantially strengthened the manuscript. Below we provide detailed itemized responses to the Reviewer concerns. Changes in the revised manuscript are made in the editor mode.
General comment
Filatov et al performed haemostatic and hematological measurements comparing Acomys cahirinus (Spiny mouse) with normal BALB/c mice (Mus musculus), and found a significantly-reduced bleeding time with a 40% higher level of blood fibrinogen in Acomys mice. This study provides important information to the existing literature helping to establish Acomys mice as a new model for the study of human physiology and pathology, especially in the area of regenerative medicine. Authors may wish to add more observations to further improve the report.
Response
We thank the Reviewer for finding our study useful and providing important information to the existing literature.
Comment 1
Data presented in Figures 1 and 2 can be combined to reduce total number of figures in the manuscript. More importantly, the argument authors made at line 348-350 in methods regarding animal gender selection is not valid. It is very important to know whether any specific change in blood counts and haemostatic measurements is gender specific, especially when Acomys mice are gaining popularity as model animals for the study of human pathology/disease therapy. Authors need to make new measurements comparing female Acomys and BALB/c mice and add the new data to current figures.
Response
Per the Reviewer’s suggestion, we have combined Figures 1 and 2, and the total number of illustrations in the manuscript went down to 6.
To address the Reviewer’s concern regarding animal gender selection, we conducted an additional experiment on the complete blood count in female Acomys cahirinus and Balb/c mice. The results indicate that the difference in cellular blood composition between Acomys cahirinus and Balb/c mice was mostly similar in female and male animals. Namely, hemoglobin and hematocrit levels in female Balb/c mice were found to be significantly higher compared to those in Acomys samples irrespective of the gender. Other parameters, such as red blood cell count (RBC), mean corpuscular volume (MCV), mean cell hemoglobin (MCH), mean cell hemoglobin concentration (MCHC) as well as platelet counts and mean platelet volume (MPV) showed no significant differences between male and female mice. The new results obtained strongly suggest that the animal gender is not crucial for the findings described in our study. In response to the Reviewer’s comment, the new results on female mice have been added to Table 1 with a corresponding description in the text on page 7. The new paragraph reads as follows.
“Haematological parameters are shown in Table 1. The mean count of red blood cells (RBC) in the blood from both male and female Acomys cahirinus was reduced compared to Balb/c, while the median mean cell volume (MCV), the mean content of haemoglobin per one erythrocyte (MCH) and the mean corpuscular haemoglobin concentration (MCHC) were significantly higher. Haemoglobin concentration and haematocrit were found to be elevated only in the samples from female Balb/c mice. Platelet counts did not exhibit a statistically significant difference between the samples obtained from both genders of Acomys, however, mean platelet volumes (MPVs) were higher in Acomys, regardless of the gender. The total white blood cells WBC count was higher in Acomys compared to Balb/c mice solely in males. Concurrently, the samples obtained from male Acomys showed an elevated lymphocyte count and a reduced neutrophil count in comparison to the Balb/c samples. Based on the results of the leukocyte counts, the neutrophil to lymphocyte ratio (NLR) was calculated, which was higher in the Acomys samples (Table 1)”.
Although animal gender may potentially affect the results, there is an important argument for perfoming the study on male mice. It is the unique (for rodents) reproductive physiology of Acomys, which, unlike Balb/c mice, exhibit a menstrual cycle similar to that of humans, implying fluctuations of the ovarian function and uterine bleeding. This important physiological peculiarity is a potential source of data variability, complicating the acquisition of precise and reproducible results. This point has been emphasized in Discussion on page 9 of the revised manuscript as follows.
“It is important to note that hematologic profiles in mice vary significantly across different studies reported in the literature [19-22], including hematocrit (HCT), mean corpuscular volume (MCV), and white blood cell (WBC) values. The differences observed in parameters such as a higher haemoglobin concentration and hematocrit among female Balb/c mice can be attributed to the fact that Acomys cahirinus, the only rodent species with a menstrual cycle, exhibits uterine bleeding associated with anemia [23]. To avoid data variability due to the ovarian and uterine cycles in female Acomys, the study of hemostatic parameters was performed in male mice.”
Comment 2
Blood collection and complete blood count were briefly described in the methods section. However, data presented in Table 1 seemed to be somewhat out of the ordinary: HCT and MCV values were too high while WBC value was too low for Balab/c mice. Were the Balb/c mice normal/healthy? Did authors make good calibration to the CBC analyzer before actual sample measurements? It is recommended that authors calibrate their CBC analyzer with good Mus musculus standards before future sample analyses.
Response
We thank the Reviewer for the insightful comments regarding the blood collection and complete blood count (CBC) data presented in Table 1. We would like to clarify that the blood analysis was conducted in a veterinary clinic using an automated hematologycal analyzer (Urit-5380 Vet, URIT Medical Electronic Group Co, China) that was calibrated for mice according to the manufacturer's instructions. It is important to note that hematologic profiles in mice vary significantly across different studies reported in the literature [19-22], including hematocrit (HCT), mean corpuscular volume (MCV), and white blood cell (WBC) values. In addition to the automated blood analysis, the number of WBCs was determined microscopically in a hemocytometer, and the obtained NLR values and WBC counts in Balb/C mice correspond to the normal values presented in the literature. These parameters are sensitive indicators of inflammation and stress, and their normal values in our study suggest the absence of underlying pathology in both mice colonies. It is noteworthy that in our study the conclusions are based on the comparative data between Acomys and Balb/c mice rather than on the absolute values of the hematological parameters.
Comment 3
The discussion at lines 269-277 regarding large platelets from Acomys mice were inaccurate or invalid since authors did not present any data concerning platelet maturity (reticular platelets) in the current study. The related discussion is baseless. According to data presented in Table 1, MPV was 10.8 for Acomys mice and was 9.2 (this MPV value is too large for Balab/c mice under normal circumstance), which represents a 17% higher MPV in Acomys mice. Since Acomys mice are generally larger and live longer than Balb/c mice, having slightly higher MPV and MCV should be regarded as normal, unless authors can provide any specific evidence showing that the platelets from Acomys mice were pre-mature.
Response
We agree with the Reviewer that without specific data on platelet maturity (reticular platelets), our previous discussion about platelet size differences between Acomys and BALB/c mice was largely speculative and not supported by our current dataset. In response to the Reviewer’s comment, the text regarding large platelets from Acomys mice has been removed from the Discussion section.
Comment 4
The Spiny mouse’s unique characteristic of “scarless wound healing” relative to normal Mus musculus mice has been the focus of the study. However, data obtained from current study indicated that Spiny mice are hyperfibriogenemia, with specific data showing in Figure 3E with larger fibrin aggregates in blood clots. Authors need to make a better connection/explanation between the “scarless” nature of Acomys mice and current observations.
Response
We appreciate the Reviewer's suggestion to emphasize the connection between the scarless wound healing observed in Spiny mice and the hyperfibrinogenemia revealed in our study. In response to the Reviewer’s comment, the following text has been added to Discussion on page 10 (lines 326-336) of the revised manuscript.
“In addition to variations in systemic inflammation, a degree of local inflammation at the site of injury may play a crucial role in this unique regenerative ability. Hyperfibrinogenemia in Acomys leads to deposition of a larger amount of dense fibrin, which is additionally compacted during enhanced clot contraction (Figs. 2 and 3). Such fibrin-rich and dense blood clot can form an impermeable seal that impedes penetration and accumulation of inflammatory cells, neutrophils and monocytes, thus impeding the local inflammation. This reduced number of immune cells in the wound leads to the lower local concentrations of inflammatory mediators, including chemotactic cytokines, followed by decreased migration and low activation of fibroblasts synthesizing collagen, the major component of a scar [36-40]. To test this presumptive explanation, additional studies examining the local cellular composition of the wound during healing dynamics are necessary”.
Reviewer 2 Report
Comments and Suggestions for Authors
The article entitled “Distinct Haemostasis and Blood Composition in Spiny Mouse
Acomys cahirinus” is a comparative study about the hemostatic system of two mammalian models such as Acomys cahirinus and Mus musculus. Acomys cahirinus Acomys cahirinus is an emerging mammalian model system capable of wound healing response and organ regeneration without fibrotic scars and loss of organ function. This characteristic can translate in rigernerative medicine and studies about the underlying mechanisms may provide useful informations. This study show that Acomys cahirinus have a hyoercoagulability state compared to Mus musculus charactyerized by high fibrinogen, blood clots denser and stiffer, time bleeding time shorter, blood clots contraction stronger and faster, and platelet hyperaggregability in terms of increased kinetic. In relation to cellular blood composition there was a reduction of RBC in association with high MCV, MCH and MCHC, high MPV, high WBC together with elevated lymphocytes and reduced neutrophils. The project study is well conducted and the used laboratory tests are appropriate with the aim of the study. In the discussion the authors provide an explanation point to point of each obtained results. However, about the RBC and the erythrocyte parameters an additional explanation could underlie hemolysis and, therefore, the reticulocyte count could be useful. In addition, the authors have chosen the MPV parameter as a indicator of activated platelets focused on the fact that platelets are large and immature, but there is also the PDW (Platelet Distribution Width) parameter, another specific marker of activated platelets focused on the fact that platelet activation causes morphologic changes of platelets, including both the spherical shape and pseudopodia formation. Platelets with increased number and size of pseudopodia differ in size, possibly affecting platelet distribution width (PDW) (Vagdatli E et al, Hippocratia 2010. I suggest adding this additional notes and the reference in the section “Discussion” with the order to provide an exhaustive information. Therefore, I think that this article is suitable for publication in a revised version.
Author Response
We would like to thank the Reviewer for their thorough review of our manuscript. To address the Reviewer concerns and excellent suggestions, we have performed additional experiments and revised the text based on the Reviewer comments. We think that these changes have substantially strengthened the manuscript. Below we provide detailed itemized responses to the Reviewer concerns. Changes in the revised manuscript are made in the editor mode.
General comment
Acomys cahirinus” is a comparative study about the hemostatic system of two mammalian models such as Acomys cahirinus and Mus musculus. Acomys cahirinus Acomys cahirinus is an emerging mammalian model system capable of wound healing response and organ regeneration without fibrotic scars and loss of organ function. This characteristic can translate in rigernerative medicine and studies about the underlying mechanisms may provide useful informations. This study show that Acomys cahirinus have a hyoercoagulability state compared to Mus musculus charactyerized by high fibrinogen, blood clots denser and stiffer, time bleeding time shorter, blood clots contraction stronger and faster, and platelet hyperaggregability in terms of increased kinetic. In relation to cellular blood composition there was a reduction of RBC in association with high MCV, MCH and MCHC, high MPV, high WBC together with elevated lymphocytes and reduced neutrophils. The project study is well conducted and the used laboratory tests are appropriate with the aim of the study. In the discussion the authors provide an explanation point to point of each obtained results.
Response
We thank the Reviewer for finding our study well conducted with appropriate methods.
Comment 1
However, about the RBC and the erythrocyte parameters an additional explanation could underlie hemolysis and, therefore, the reticulocyte count could be useful. In addition, the authors have chosen the MPV parameter as a indicator of activated platelets focused on the fact that platelets are large and immature, but there is also the PDW (Platelet Distribution Width) parameter, another specific marker of activated platelets focused on the fact that platelet activation causes morphologic changes of platelets, including both the spherical shape and pseudopodia formation. Platelets with increased number and size of pseudopodia differ in size, possibly affecting platelet distribution width (PDW) (Vagdatli E et al, Hippocratia 2010. I suggest adding this additional notes and the reference in the section “Discussion” with the order to provide an exhaustive information. Therefore, I think that this article is suitable for publication in a revised version.
Response
We thank the Reviewer for the valuable and positive feedback on our manuscript. We appreciate the Reviewer’s insights, particularly regarding the RBC and platelet parameters. A similar comment regarding MPV was made by other Reviewer, and we agree that this parameter is not sufficient by itself to draw the conclusion about continuous platelet activation. In response to the concerns presented by both Reviewers, we have removed this paragraph from the Discussion section. We believe this minor change will enhance the clarity and focus of our manuscript.
Round 2
Reviewer 1 Report
Comments and Suggestions for Authors
The revised manuscript showed improvements as authors added female CBC measurements to Table 1. Very good! However, authors did not address the issue of adding females to other haemostatic measurements as previously requested.
In the responses authors stated that “The new results obtained strongly suggest that the animal gender is not crucial for the findings described in our study”. Yet in methods authors made a contradictory argument that: “The selection of the male animals was made to avoid any potential variability in the study results that could arise from sex-related factors, such as the menstrual cycle and the fluctuation in hormone concentration, which may confound the interpretation of the data” (lines 368-371). This argument along with some of the discussions/description authors presented in the revised manuscript showed obvious gender bias in this study.
In the responses authors stated that: “Although animal gender may potentially affect the results, there is an important argument for performing the study on male mice. It is the unique (for rodents) reproductive physiology of Acomys, which, unlike Balb/c mice, exhibit a menstrual cycle similar to that of humans, implying fluctuations of the ovarian function and uterine bleeding. This important physiological peculiarity is a potential source of data variability, complicating the acquisition of precise and reproducible results. This point has been emphasized in Discussion on page 9 of the revised manuscript as follows”.
Authors presented a very good argument here but this argument should provide strong justification for the inclusion of females, not exclusion of females, in any investigation using Spiny mice as model animals for the study of human physiology and pathology!
Author Response
Response to the Reviewer’s comment on the manuscript by Filatov et al. entitled “Distinct Haemostasis and Blood Composition in Spiny Mouse Acomys cahirinus”
(Manuscript ID: ijms-3291251)
We sincerely appreciate the Reviewer’s suggestion to add female mice to our study, as it has enriched our findings and generalized understanding of the haemostatic peculiarities in Acomys cahirinus. We think that these changes have substantially strengthened the manuscript. We respond in detail to all the reviewer concerns below, and the changes in the manuscript are indicated by the use of red font.
Reviewer’s comment
The revised manuscript showed improvements as authors added female CBC measurements to Table 1. Very good! However, authors did not address the issue of adding females to other haemostatic measurements as previously requested.
In the responses authors stated that “The new results obtained strongly suggest that the animal gender is not crucial for the findings described in our study”. Yet in methods authors made a contradictory argument that: “The selection of the male animals was made to avoid any potential variability in the study results that could arise from sex-related factors, such as the menstrual cycle and the fluctuation in hormone concentration, which may confound the interpretation of the data” (lines 368-371). This argument along with some of the discussions/description authors presented in the revised manuscript showed obvious gender bias in this study.
In the responses authors stated that: “Although animal gender may potentially affect the results, there is an important argument for performing the study on male mice. It is the unique (for rodents) reproductive physiology of Acomys, which, unlike Balb/c mice, exhibit a menstrual cycle similar to that of humans, implying fluctuations of the ovarian function and uterine bleeding. This important physiological peculiarity is a potential source of data variability, complicating the acquisition of precise and reproducible results. This point has been emphasized in Discussion on page 9 of the revised manuscript as follows”.
Authors presented a very good argument here but this argument should provide strong justification for the inclusion of females, not exclusion of females, in any investigation using Spiny mice as model animals for the study of human physiology and pathology!
Response
In response to the Reviewer’s insightful comments regarding the inclusion of female animals in our study, we have conducted additional experiments on female subjects using the same techniques applied originally to the male cohort. Importantly, the new results obtained in female mice have confirmed our previous major conclusions based on the male mice data, indicating that the difference in haemostasis between Acomys and Balb/c mice are mostly gender-independent and have fundamental biological nature. The results obtained from these additional experiments have been meticulously analyzed and included in the revised manuscript. We have updated the relevant sections accordingly to reflect these findings, ensuring that our conclusions are based on a balanced representation of both genders. This enhancement not only addresses the concerns raised by the Reviewer about potential gender bias but also strengthens the overall validity of our research. We believe that these changes will provide a more accurate and inclusive perspective on the distinct haemostasis and blood composition in spiny mouse Acomys cahirinus.
Round 3
Reviewer 1 Report
Comments and Suggestions for Authors
Data from females were added in Figures 1,3, 4, and 5 that enhanced the study significantly. The different responses from male and female Acomys cahirinus mice, relatively to Balb/c mice, illustrated the importance of using both males and females in any specific study involving animal models! The arguments authors made in discussion (lines 312 to 319) in reference to previous publications are acceptable.